# The Effects of Heavy Metal Pollution on Collembola in Urban Soils and Associated Recovery Using Biochar Remediation: A Review

**DOI:** 10.3390/ijerph20043077

**Published:** 2023-02-09

**Authors:** Alexis Kayiranga, Zhu Li, Alain Isabwe, Xin Ke, Claudien Habimana Simbi, Binessi Edouard Ifon, Haifeng Yao, Bin Wang, Xin Sun

**Affiliations:** 1Key Laboratory of Urban Environment and Health, Ningbo Observation and Research Station, Institute of Urban Environment, Chinese Academy of Sciences, Xiamen 361021, China; 2Zhejiang Key Laboratory of Urban Environmental Processes and Pollution Control, CAS Haixi Industrial Technology Innovation Center in Beilun, Ningbo 315830, China; 3University of Chinese Academy of Sciences, Beijing 100049, China; 4CAS Key Laboratory of Soil Environment and Pollution Remediation, Institute of Soil Science, Chinese Academy of Sciences, Nanjing 210008, China; 5Center for Excellence in Molecular Plant Sciences, Chinese Academy of Sciences, Shanghai 200032, China; 6College of Earth and Planetary Sciences, University of Chinese Academy of Sciences, Beijing 100049, China

**Keywords:** collembola, urban soil, heavy metals, biochar remediation

## Abstract

Heavy metal pollution in urban soil continues to be a global issue that poses a serious hazard to invertebrates and human lives through oral ingestion and inhalation of soil particles. Though the toxicity of several heavy metals on invertebrates like Collembola has been studied, lead (Pb) and cadmium (Cd) have been extensively studied due to their high toxicity to collembolans. As a ubiquitous soil organism all over the world, collembolans have been used as a model species to study the effects of heavy metals on invertebrate communities. To reduce the effects of heavy metals on ecosystem functions, biotic and abiotic measures have been used for heavy metal remediation; biochar seems to be the most effective approach that not only increases the physical absorption of heavy metals but also indirectly benefits soil organisms. In this study, we briefly reviewed the application of biochar in Pb and Cd polluted soil and showed its potential in soil remediation. Furthermore, we outlined the potentially toxic effects of Pb- and Cd-polluted urban soil on the collembolan species. We searched peer-reviewed publications that investigated: (1) the level of Pb and Cd contamination on urban soil in different cities around the world; and (2) the different sources of Pb and Cd as well as factors influencing their toxicity to collembolan communities. The obtained information offers new perspectives on the interactions and effects between collembolans, Pb, and Cd, and their remediation in urban soils.

## 1. Introduction

Urban soil is polluted by the accumulation of heavy metals during the rapidly expanding urbanization and industrialization processes of the Anthropocene [1,2,3]. However, based on over-exploitation and insufficient remediation technology, metal processing and mining activities are estimated to account for 48% of total pollutants released by the European industrial sector [4,5]. Soil pollution by heavy metals has become a serious worldwide problem, especially in urban areas due to urbanization and industrialization [6]. A number of ecosystems near urban areas in many countries are polluted by heavy metals, mostly lead (Pb) and cadmium (Cd) [7,8,9,10]. For instance, more than 13% of the total cultivated land, about 40% of the lakes and rivers, and around 0.24 billion hectares of the world’s arable land are polluted by heavy metals [8,11,12,13]. Heavy metals are a threat to soil and ecosystem health because they can stay in the soil for a long period of time and they are toxic to most soil organisms that drive most of the ecological processes [14,15,16]. Consequently, human health is at risk from famine and diseases associated with heavy metal pollution. For example, in Japan, 2252 and 1043 people were affected and died due to Minamata disease caused by heavy metal contamination, respectively [17].

Pb and Cd soil pollution in urban soils has become a serious issue in China as a result of rapid urbanization and industrialization over the last two decades, similar to developed countries in America and Europe. Studies conducted by Li and Wei et al. showed that, in China, the levels of contamination in the geological background are low [18,19]. However, the same studies reported that China’s urban soil environments are highly polluted by Pb and Cd. Excessive Pb and Cd pollution endangers the health of not only humans but also soil invertebrates (e.g., collembolans). For example, collembolans have been shown to be highly susceptible to heavy metal pollution, which they take directly or through food [20,21,22,23,24]. Wei and Yang also reported that the accumulation of Pb and Cd in soils, especially those in the vicinity of industrial areas, is released from anthropogenic activities [19]. Therefore, the ever-increasing Pb and Cd pollution in urban soil necessitates the exploration of the impact of these heavy metals on soil ecosystems [25]. Urban soils are generally considered a complex terrestrial ecosystem because of their variety of physicochemical properties. This is because soils in urban areas are generally refilled and transported from nearby systems, lake settlings, deep soils, and even construction residues.

Soil organisms (e.g., bacteria, fungi, protists, nematodes, and soil fauna) play a prominent role in driving soil ecosystem services and functions, as they are involved in energy fluxes and nutrient cycling [26]. Among the soil-dwelling arthropods, collembolans are the most common and abundant arthropods and have a wide range of dietary spectrum and spatial distribution [27,28,29,30,31,32]. More than 50 species of collembolan can be found in 1 m^2^ of natural forest and grassland soils; overall abundance often exceeds 10,000 m^2^ and can reach 60,000 m^2^ [33]. Collembolan diversity is linked to a wide range of living forms as well as specialized ecological adaptations. In contrast to natural sites, collembolan richness and abundance are much lower in agricultural landscapes [34]. This decrease has been linked to higher concentrations of pollutants and lower resource availability [31]. The decrease in collembolans in agricultural landscapes can negatively affect nutrient cycling, resulting in a decrease in agricultural productivity [29,35].

Environmental changes are frequently connected to changes in the species composition of collembolans [31,32]. Studies have shown that collembolan species respond quickly and sensitively to environmental changes and disturbances, particularly in the presence of widespread pollutants [26,31,36]. Therefore, collembolans are considered bio-indicators of soil pollution as well as soil fertility due to their wide distribution and sensitivity to pollutants [28]. Additionally, Pb and Cd have been reported to have a profound effect on collembolan species structure and living behaviors due to their sensitivity to pollutants [37]. Consequently, the pollutants in urban soils released from anthropogenic activities cause more stress on the collembolan species since they may have different effects on collembolan life-cycle characteristics [26,32,38]. This implies that an exceedingly high percentage of Pb and Cd pollution in urban soil can be transferred to higher trophic level organisms through the food web from collembolans and could affect the stability of soil ecosystem species [31]. Pb is a non-essential metal that is challenging for collembolans to control once ingested [32]. Therefore, a collembolan’s internal Pb concentration may be a reliable indicator of both its bioavailable fraction and possible risk.

Though the impact of heavy metal pollution on collembolans in the field has received a lot of attention, the majority of studies have been undertaken in Europe [33,34], with only a few being conducted in China [39]. Following the dearth of similar studies in China and other parts of the world, recent studies have recommended more research on the fate and toxicity of Pb and Cd and their relative conditions of exposure [40,41]. However, such studies are difficult because the assessment of the threats of Pb and Cd in soil organisms is more difficult than that of aquatic animals [41]. The difficulties arise from the fact that the interaction between collembolans and heavy metals is more complex. Conducting a full study on the effect of Pb and Cd on the collembolan species involves evaluating the level at which collembolans can be polluted and examining the threats of Pb and Cd at the individual, population, and community levels. Consequently, studies examining the toxicity of urban soil pollutants and their effect on soil organisms like collembolans are of great importance in urban risk assessment.

Meanwhile, it is of critical importance to fully study and comprehend the behavior of Pb and Cd in urban soils and its appropriate remediation [42,43,44]. Several techniques and technologies for Pb and Cd removal from the soil environment have been reported [45] and reviewed [44,46]. To the best of our knowledge, most research has concentrated on wastewater and expensive technology that might have a negative environmental impact. Recently, it has been suggested that using biochar, a solid carbonaceous substance, is an efficient green method for immobilizing hazardous pollutants in water and urban soil such that they are not readily available to collembolans [44,47]. Biochar is a promising alternative amendment that is more efficient for positively charged heavy metals (e.g., Pb and Cd) because of its negative charge, low cost, high efficiency, and environmentally favorable qualities [44,47]. However, predicting which technologies from a large variety of platforms will be more efficient in producing specific biochar with specified qualities for use in Pb and Cd remediation is quite difficult. The effectiveness of using biochar as an approach to remediate Pb and Cd can affect nutrient cycling in three ways: (i) as a direct supply of nutrients; (ii) by encouraging collembolan species activity, which speeds up decomposition rates and mineralization by grazing on *fungi* or *humus*; and (iii) by preventing nutrient loss. Surprisingly, biochar has been shown to minimize nutrient loss by leaching, notably by enhancing the retention of cationic ions [48]. In general, the lower the nutrient content of the soil, the more detrimental the effect on collembolan reproduction at a given concentration [49].

Therefore, this review aims to provide a better understanding of the effects of Pb and Cd pollution on collembolan species. Among the collembolan species in this review, we focus on *Folsomia candida* and *Coecobrya caeca* and demonstrate their sensitivity to Pb and Cd using laboratory tests. We also summarized the major sources and compared the levels of Pb- and Cd-contaminated urban soil in cities in China and other countries. Thus, we discussed the interactions between Pb and Cd and collembolan species, the fate of Pb and Cd on urban soil, and the current technology to use for the response characteristics to urban Pb and Cd pollution.

## 2. Heavy Metal Contamination in Urban Soils from Around the World

### 2.1. Levels of Pb and Cd Pollution in the Urban Soil

For pollution control and urban planning, studies on contamination levels and regional variations of trace metals in urban soils are crucial. In this analysis, we analyzed data from numerous studies that were published between 1994 and 2021 to determine the amounts of Pb and Cd pollution in urban soils in specific cities around the world. A descriptive statistic of Pb and Cd in urban soils in the chosen Chinese cities is given in Figure 1 and Appendix A due to the possible contribution of Pb and Cd in the urban region to relatively higher pollution of urban soils globally. For example, in all the cities in China, the mean levels of Pb and Cd in urban soils were observed at 1350.51 mg kg^−1^ (ranging from 28.6 to 25,380.55 mg kg^−1^) and 1.58 mg kg^−1^ (0.15–8.59 mg kg^−1^), respectively [19]. In Shanghai, China, a particularly high mean of 70.69 ± 5.09 and 0.52 ± 0.05 mg kg^−1^, for Pb and Cd, respectively, were reported from a total of 273 sampling sites within the city [6,50]. These reported Pb and Cd contamination levels in areas like Shanghai are a result of China’s fast industrialization and urbanization after the implementation of its open-door policy in 1978. The establishment of numerous foreign businesses in Shanghai, including electroplating, chemical, and electronic and electrical companies contributed to the city’s quick economic growth. These industries in Shanghai have been discharging large amounts of Pb and Cd for the last 20 years. The mean concentrations of 93.40 ± 37.30 (5.27–404.00) mg kg^−1^ for Pb and 2.18 ± 1.02 (0.02–5.89) mg kg^−1^ for Cd were identified from 594 samples from 65 urban soil parks in Hongkong, and this is the most comprehensive investigation of Pb and Cd concentrations in urban soil to date [51].

In addition, the surface soils of the Guangzhou urban area appeared to be polluted by Pb and Cd to some extent. In summary, the total concentrations of these metals ranged from 18.50 to 348.40 mg kg^−1^ for Pb and 0.03 to 2.41 mg kg^−1^ for Cd. The soil samples used were from residential sites, roadside areas, industrial sites, and urban parks. The average Pb and Cd levels detected from these study sites were 108.55 ± 78.28 and 0.49 ± 0.49 mg kg^−1^, respectively [52]. Based on the results above, when compared with the adjoining cities of Shanghai and Hongkong, the contamination of Pb in Guangzhou is much higher than that observed in Shanghai and Hongkong, but that of Cd is quite similar. In Hangzhou (China), a range of 54.00 to 845.00 mg kg^−1^ (Pb) and 0.29 to 5.70 mg kg^−1^ (Cd) were obtained from 25 polluted road-deposited sediments in five different land-use zones (industrial, residential, commercial, park, and countryside), with a mean of 202.16 ± 197.62 and 1.59 ± 1.41 mg kg^−1^, respectively. The highest metal concentration levels in Hangzhou were detected in the samples from the industrial and commercial zones experiencing heavy traffic [42,53]. Furthermore, the urban and suburban soil samples from Lishui city showed a range of Pb = 29.18 to 166.93 mg kg^−1^ with an average of 63.15 ± 21.96 mg kg^−1^ and Cd = 0.05 to 1.90 mg kg^−1^ with an average of 0.53 ± 0.52 mg kg^−1^ observed in the urban and suburban soils of Lishui city samples [54]. Furthermore, in Changchun, China, the means of Pb and Cd levels were detected as 35.4 ± 26.3 mg kg^−1^ and 0.13 ± 0.59 mg kg^−1^, respectively, in which a total of 352 samples of urban soil were collected from urban areas of Changchun using a systematic sampling strategy in which one sample per km^2^ was taken in the uppermost (0–20 cm) contaminated urban soil [55,56].

Urbanization is high in most European countries, and approximately 80% of the population in Europe lives in urban areas. The high levels of urbanization increase Pb and Cd contamination in these urban soils, especially due to high traffic emissions and industrialization in these areas [56,57]. In comparison to other countries, Pb and Cd concentrations in 16 urban gardens in Salamanca province (Spain) ranged between 20.10 and 96.20 mg kg^−1^ (on average, 53.10 mg kg^−1^) and 0.20–0.95 mg kg^−1^ (0.53 mg kg^−1^), respectively, due to Pb and Cd polluted urban soils [58]. The Thane district of Maharashtra, India, reported Pb and Cd content ranging from 35.9 to 49.20 and 21.60 to 38.60 mg kg^−1^, respectively, with Swedish guideline values (SGV) for levels in polluted soils, whereas Pb was 80 mg kg^−1^ and Cd was 0.40 mg kg^−1^. This pollution could be caused by the discharge of industrial waste or leaching from sewage-laden landfills. The district has abundant natural resources in the form of perennial rivers, an extensive seashore, and a high mountain range [59]. The mean Pb and Cd contents of 60.20 ± 10.30 mg kg^−1^ (20.50–117.00 mg kg^−1^) and 6.55 ± 1.29 mg kg^−1^ (2.50–1.50 mg kg^−1^), respectively, were detected in urban soil around Queen Alia Airport, Jordan, at a depth of 0–10 cm [60]. This pollution was mainly caused by basaltic rocks and, anthropogenic rocks. Pb concentrations in topsoil (0–10 cm) ranged from 8.00 to 129.00 mg kg^−1^ and mean 35.20 ± 43.30 mg kg^−1^, while the highest level of 129.00 mg kg^−1^ was observed in the industrial area, the lowest level of 8.00 mg kg^−1^ was detected in a residential and commercial area of Talcahuano, Chile (South America); one factor contributing to the greater concentration of Pb in the topsoil, along with the probable pollution, was the metal’s affinity for organic matter and its limited mobility [61]. Cd detected in this country was below the 0.25 mg kg^−1^ detection limit. For this reason, this metal was not presented in Table 1. Urban soil pollution is influenced by the age of the city, land use, population density, environmental laws, and regional climate. The “urban” component, which incorporates the intensity and kind of contaminant emissions and anthropogenic disturbances, seemed to prevail in determining trace metal pollution in urban soils, regardless of the location, climate, and size of cities.

The majority of the samples from cities in Europe and America, in contrast to Nigeria in Africa, and Thailand, China, Mongolia, and Turkey in Asia, were moderately or seriously polluted, with Pb and Cd considerably contributing to the pollution. Despite the fact that pollution is often low in the developing world (Table 1) [56,62], there are also instances of highly contaminated urban soil, such as in the industrial or newly growing megacities of Sialkot and Islamabad in Pakistan and Moscow in Russia. Additionally, in both of Naples’s urban districts, a general decline in Pb and Cd concentration was found 10 to 15 years ago [56]. For instance, Luo and Milenkovic reported that Pb and Cd data concentrations were observed in 9,954 samples from 34 European cities and found the following mean values [56,63,64,65]: Pb was 102.00 mg kg^−1^ and Cd was 0.95 mg kg^−1^, while a mean of 231.00 mg kg^−1^ of Pb and 1.10 mg kg^−1^ of Cd were detected from 122 soil contaminated samples taken in Baltimore (USA); Chicago exhibited 395.00 mg kg^−1^ of Pd from 57 samples, but, fortunately, the amount of Cd detected in this city was below the detected limits [66]. Furthermore, 47.80 ± 52.70 mg kg^−1^ (12.10–269.30 mg kg^−1^) and 0.34 ± 0.21 mg kg^−1^ (0.05–0.81 mg kg^−1^) for Pb and Cd, respectively, were reported in the continental crust and urban soils in Bangkok (Thailand), while Moscow (Russia) indicated mean concentrations of Pb = 37.00 mg kg^−1^ and Cd = 2.00 mg kg^−1^. In Islamabad (Pakistan), Pb = 212.34 ± 1.87 mg kg^−1^ and Cd = 3.54 ± 0.06 mg kg^−1^ were detected in built-up areas. In Sialkot (Pakistan), the mean concentrations of Pb = 121.40 ± 18.05 mg kg^−1^ and Cd = 36.80 ± 11.99 mg kg^−1^ were observed from 82 samples. Mean Cd concentration exceeded toxic limits of 3.00 mg kg^−1^ for urban soils [67,68,69,70]. In Izmit, an industrial city in northern Turkey, a mean of 32.00 ± 11.00 mg kg^−1^ (8.00–45.00 mg kg^−1^) and 0.21 ± 0.09 mg kg^−1^ (0.07–0.35 mg kg^−1^) of Pb and Cd, respectively, were observed from 28 polluted continental crusts.

Notably, the mean concentrations of Pb = 63.90 mg kg^−1^ (22.57–119.62 mg kg^−1^) and Cd = 0.80 mg kg^−1^ (0.43–1.10 mg/kg) were detected in Ulaanbaatar (Mongolia) [71]. These results showed that Pb pollution is related to vehicle emissions resulting from the use of leaded gasoline [72]. Recently, the number of used cars in Ulaanbaatar has increased sharply, and used cars release a considerable amount of smoke containing Pb and other toxic chemicals. Unfortunately, leaded gasoline is still being used in Mongolia. Rapid urbanization has taken place in the city over the past 10 years. In general, metal pollution was not a severe issue in the city, and there was little proof that metal solutions had seeped into the soil. However, the notable increase in old car sales and the rise in leaded fuel consumption over the past few years may be the main causes of the increase in Pb pollution.

**Table 1 ijerph-20-03077-t001:** Mean concentrations (mg kg^−1^) of Pb and Cd in urban soils from some cities of the world in comparison to China.

Location	Sample	Pb		Cd		References
		Mean	Range	Mean	Range	
34 European cities	9954	102.00	-	0.95	-	[56]
Salamanca province (Spain)	16	53.10	20.10–96.20	0.53	0.20–0.95	[58]
Maharashtra (India)	12	80.00	35.90–49.20	0.40	21.60–38.6	[59]
Queen Alia Airport (Jordan)	32	60.20 ± 10.30	20.50–117.00	6.55 ± 1.29	2.50–1.50	[60]
Talcahuano, Chile (South America)		35.20 ± 43.3	8.00–129.00	-	-	[61]
Baltimore (USA)	122	231.00	54.60–1013.70	1.1	0.54–1.83	[66]
Chicago (Chicago)	57	395.00	-	-	-	[66]
Bangkok (Thailand)	30	47.8 0 ± 52.70	12.10–269.30	0.34 ± 0.21	0.05–0.81	[67]
Moscow (Russia)	36	37.00	-	2.00	-	[68]
Islamabad (Pakistan)	307	212.34 ± 1.87	-	3.54 ± 0.06	-	[69]
Sialkot (Pakistan)	82	121.40 ± 18.05	-	36.80 ± 11.99	-	[70]
Izmit industrial city (Northern Turkey)	41	32.00 ± 11.00	8–45	0.21 ± 0.09	0.07–0.35	[72]
Ulaanbaatar (Mongolia)	22	64.00	-	0.8	-	[71]

### 2.2. The Fate of Pb and Cd in Urban Soils

Industrial processes (such as those at power plants and cement plants), mining operations, irrigation with dirty water, and weathering release from parent rocks are all potential sources of Pb and Cd pollution in urban soils [42,44,47], fly ash with Pb and Cd that has been discharged into the atmosphere by mining and industrial processes has the potential to travel great distances and deposit in adjacent ecosystems, damaging irrigation water and urban soil. Additionally, industrial effluents and outflows from mining and other industrial activities could release Pb and Cd into the environment. Large amounts of Pb and Cd are released into the water and urban soil by mining and related operations [42,73,74]. Based on its comparable ionic radius, Cd can substitute for divalent cations, including Ca, Fe, Zn, Pb, and Co, in the soil, e.g., in carbonate and phosphate rocks [75]. Herein, sulphate and hydroxide forms of Pb and Cd can coexist in dry and wet deposition (clouds, rain, fog, and snow) [44,76]. Less water-soluble Pb and Cd sulfides, sulphates, oxides, carbonates, and hydroxides can only be dispersed and distributed in the atmosphere before being deposited by gravity settling. Pb and Cd are also released into the atmosphere, primarily by internal combustion engine exhaust gases and, to a lesser extent, through smoke from large-scale industrial coal burning. Therefore, compared to rural areas, the amount of Pb released is larger in urban industrial locations [44,76,77]. Cd can substitute for Ca in apatite, which is the principal constituent of phosphates. Consequently, Cd is a common impurity in phosphate minerals and phosphoritic rocks, which are prominent for fertilizer production [75,76]. Pb and Cd compounds are not likely to volatilize into the atmosphere after they are deposited in urban soil or water, but in dry conditions, wind dispersion from the soil surface is feasible. Additionally, Pb and Cd can be partially released from the parent rocks by weathering. As a result, Pb and Cd may be absorbed and transported directly to a variety of edible plant components, endangering the health of urban residents who eat food from Pb- and Cd-contaminated plants [44,74].

## 3. Toxicity of Pb and Cd on Collembolans in Contaminated Soil and Laboratory Tests

### 3.1. Performance of Collembolans in the Field

Collembolans and the human body may be directly exposed to heavy metals in urban soil through oral ingestion, dermal contact, and inhalation of soil particles. Pb and Cd pollution could not only have an adverse effect on human health and urban agriculture in cities across the world, but also cause a detrimental effect on the collembolans. Pb and Cd result in reductions in collembolan species in the urban soil via direct acute toxicity as well as changing or contaminating their food supply (as they are very sensitive to changes in contaminated urban soil) [31]. In addition to the other signs, body length may be an indicator with considerable potential for identifying the dangers of potentially hazardous metals (Pb and Cd). Pb and Cd released into urban soil from anthropogenic activities can enter a collembolan, and their concentrations in the body coincide with the Pb and Cd concentrations in the soil, which cause a toxic impact on the collembolans and affect their abundance, distribution, and species richness (Figure 2) [24,32,78,79]. Collembolan inhabiting soils polluted by Pb and Cd, due to their behavioral characteristics as well as burrowing and feeding activities, are more sensitive to Pb and Cd pollution than other groups of terrestrial invertebrates that require oxygen for metabolism. However, the toxicity of Pb and Cd to the collembolan species is different [49,79,80]. According to several studies, collembolan reproduction and adult survival were considerably reduced following exposure to Cd- and Pb-contaminated soil, but their toxicity levels were different [78,79,80,81,82]. For example, exposure of *F. candida* and *C. caeca* to higher concentrations of Pb and Cd in urban soil may drastically reduce reproduction, adult survival, growth, sexual development, and life behavior and cause changes in the size, density, and composition of *F. candida* and *C. caeca* [24,78,79,80,83]. Additionally, Cd exposure may cause mortality in the collembolan species, which eventually causes a reduction in the collembolan population size in contaminated urban soil [30,41].

### 3.2. Performance of Collembolan Exposure Test in Laboratory

Due to their simplicity in a laboratory culture and their relatively quick generation durations at ambient temperature, collembolans have been the most frequently utilized groups for soils [82]. Most studies have used *F. candida*, leading to the publication in 1999 of a recommended protocol by the International Standards Organization (ISO) [49]. For instance, the bioassay developed in Europe using *F. candida* is sensitive and very helpful for evaluating the ecological risks of polluted soils, however, it makes use of a species that is not very relevant to Australian ecosystems. Although it has been discovered locally, *F. candida* is quite uncommon, and the other collembolan species used in these studies do not exist in Australia. Australian soils often contain less organic matter and have a lower pH than European soils, which are the source of the majority of the toxicity data [84]. For instance, OECD (Guidelines for Toxicity Test Methods for Soil Organisms) artificial soil, which was created to mimic European soil conditions and used in collembolan testing, has a higher pH and organic matter content than the majority of Australian soils. In addition, Menta et al. investigated the effects of Pb and Cd on the biology (survival and mortality) of two euedaphic collembolan species, *F. candida* and *C. caeca*, as well as differences in sensitivity between the two species [80]. Thus, the two species were used in this study as the indicator species, and the soil concentrations of both elements used in this study were 10, 50, 100, 500, and 1000 mg kg^−1^ in dry soil, but no tests using both metals in one treatment were carried out. Particularly, individual survival of *C. caeca* adults was not affected by Pb concentrations of 10 mg kg^−1^ (*p* > 0.05), but individual mortality increased with concentrations of 50, 100, and 500 mg kg^−1^ (*p* < 0.01 for all concentrations). Cd caused a significant reduction in the survival of *C. caeca* adults at all concentrations tested [80]. On the other hand, there was little indication of Pb in the adult *F. candida* population’s survival. In fact, a slight decrease (weak effect) in adult survival was found in all Pb concentrations 10 days following the start of the experiment, but the differences were not significant when compared to the control (*p* > 0.05).

In another study conducted by Ding et al., a single species test was conducted using the collembolan *F. candida* as the indicator species and soil Pb pollution concentrations (0, 300, 600, 1200, 2400, and 4800 mg kg^−1^) [79]. The results from this study exhibited that adult survival started to decrease at soil Pb concentrations of 1200 mg kg^−1^ indicating that Pb did not show a strong effect on the tested species. Generally, this implies that the collembolans are either sensitive or tolerant to soil contamination, integrating both direct and indirect metal effects [39,49]. Only the effects of Pb and Cd on *F. candida* have been directly measured. According to most research, *F. candida* is one of the most Pb- and Cd-sensitive springtails compared with other collembolan species. The cause of this sensitivity is that *F. candida* is a widespread and common animal present in soil [49]. Another species that was a good test species was *Proisotoma minuta*, which can produce more than 100 eggs in just 28 days [84]. The eggs were placed in clusters, making it simple to separate them. However, counting them was challenging due to their diminutive size and light gray color. This species could become much more desirable as a test species with advancements in counting methods, such as the use of image analysis. The reproduction rates of the other species in the culture were insufficient to meet the test’s acceptability requirements (>100 instars per 28 days). *Sinella communis* and *P. minuta* are also two ecologically significant species that can be used to evaluate the toxicity of Australian soils [84]. They both have adequate rates of reproduction in artificial environments and are sensitive to a variety of toxins.

## 4. Factors Influencing the Toxicity of Pb and Cd to Collembolan Species

### 4.1. Interaction between Collembolans, Pb, and Cd

Plant detritus, animal residues, and excreta deposited on the soil surface and below are sources of organic matter in soils. The colonization of detrital matter is dominated by collembolans, which feed on the oxidative breakdown of complex organic compounds. Collembolans are vital in this process because they stimulate microbial activity, aerate and mix the soil, and improve organic matter turnover. Collembolans have a high capacity for accumulating hazardous substances; however, the extent of accumulation is determined by the type of element and soil parameters. Pb- and Cd-rich metalliferous soils are known to be home to collembolans [80]. They are likely to acquire pollutants present in soils due to their close interaction with them through eating and skin contact in both the solid and watery phases (Figure 2) [85,86,87,88]. According to numerous papers, it is often hard to directly compare the toxicity of Pb and Cd. The main justification is that while studying the toxicity of Pb and Cd on collembolan species in urban soil, both the total concentration and the bioavailable fractions should be taken into account (Figure 2) [31,32]. The latter was quite easy for collembolans to absorb through ingestion and soft skin. The bioavailable fraction is thought to be a consequence of various complex factors, including multiple chemical forms of the same metal, soil physicochemical features, and aging/leaching treatments of soil that has just been spiked with a metal compound. Additionally, test parameters such as soil moisture content, test temperature, fake food addiction, and exposure length have a significant impact on metal toxicity [88]. Although there are several evident advantages of using artificial urban soil in toxicity tests, such as the practicability and comparability of test results in different studies from an ecological overview, the use of natural soils is recommended [88]. Natural urban soils have a wide range of physicochemical features, including adsorption phases (clay, organic matter, and metal oxyhydroxides), cation exchange capacity (CEC), and soil pH [88,89]. For instance, soil acidity affects the collembolan’s survival and reproduction, and pH has a major influence on the extent of Pb and Cd effects on the collembolan community [24,90,91].

Additionally, the degree of interaction with the bioavailable percentage of Pb and Cd, as well as the behavior, distribution, and physiological function of the collembolan species, are also affected by urban soil moisture and temperature [24,92]. According to several studies, organisms spend the majority of their lives in genuine natural soil environments under suboptimal, changeable, and, occasionally, stressful environmental conditions [3,88,93]. As a result, fluctuating moisture/temperature conditions should be included in laboratory toxicity tests for better extrapolation of toxicity values in field soil. Only a few studies have investigated the effects of different moisture and temperature levels on Pb and Cd toxicity. Soil properties have been well documented as modifying metal valence state and solid-solution partitioning in soil, altering metal bioavailability, and toxicity to collembolans [32,94]. The results of toxicity studies performed on one soil type cannot be applied to another. Furthermore, soil properties contribute to the fitness and sensitivity of collembolan species [24,30,31,32,88]. For example, alkaline and low organic matter urban soils are not ideal for collembolan species reproduction. The absence of rivalry with H^+^ for metal binding increased with increasing soil pH, which was attributable to the absence of competition with H^+^ for metal binding at high soil pH [88,95,96]. On the other hand, the effect of pH on anion metals is usually the opposite of the effect of pH on cationic metals. Pb and Cd were found to be more hazardous in urban soil with a higher pH. The soil adsorption phases encourage metal sorption in the soil, lowering toxicity. However, it should be noted that determining the role of organic matter might be challenging at times. It was discovered that collembolan species were harmed when organic-matter-bound Pb and Cd was consumed [80]. The decrease in Pb and Cd toxicity was shown to be linked to an increase in soil clay [81]. The effects of metal oxides were only explained by Criel et al. who discovered that increasing amorphous manganese was substantially associated with lowering Pb and Cd toxicity [89]. Furthermore, the CEC is viewed as an integrated measurement for the number of sorption sites, taking into account clay, metal oxyhydroxides, organic matter, and pH [88,89]. Few studies have used regression analysis to construct a prediction model of Pb and Cd toxicity to collembolan species as a function of soil parameters for collembolans.

To study the effects of heavy metals on collembolan species, Pb and Cd bioavailability and toxicity are largely determined by their chemical forms. Different Pb and Cd compounds with the same ions have different physical and chemical properties, resulting in toxicity differences [24,76,97,98,99]. Therefore, toxicity based on one type of metal cannot be immediately applied to another. Because of their great solubility, metals are usually supplied as nitrate, chloride, or sulphate salts in most experiments. In reality, the examined metal’s anionic partner contributes to metal toxicity in salt-intolerant collembolan species, interfering with the display of metal toxicity impact [89,97,99,100]. For example, Owojori et al. discovered that CdSO_4_^2−^ was more toxic to collembolan species than CdCl_2_ [99]. Van et al. reported that the EC50 for reproduction and the LC50 for the survival of PbCl_2_ were found to be 3 and 3.3 times higher than those of Pb (NO_3_)_2_, respectively [100]. This is implying that a metal’s solubility is frequently needed for its toxicity to affect collembolans, and because there is no salinity stress, metal oxides and simple metal powders are generally less poisonous than their soluble metal salt counterparts.

### 4.2. Bioavailable Fractions Predicting Metal Toxicity to Collembolans

Contact with Pb and Cd via the urban soil liquid phase is a critical exposure pathway for the collembolan species. The overall Pb and Cd contents of urban soil are usually thought to be poor predictors of toxicity. Bioavailable Pb and Cd concentrations, such as porewater concentration and the extracted fraction, are directly associated with Pb and Cd toxicity and are thus thought to be more reliable in predicting Pb and Cd toxicity [101,102,103,104,105]. Nikolic et al. observed that in spiked artificial soil, the porewater concentration, the CaCl_2_-extracted fraction, and the water-extracted fraction were all good predictors of chronic Zn toxicity [103]. Similarly, when different types of soil were utilized, water-extracted Zn and porewater Zn were observed to be adequate to account for differences in toxicity thresholds. Gestel et al. reported that the LC50 values for the influence of Cd on survival differed by a larger factor between soils when expressed on the basis of water-extracted and porewater fractions, but the variation in EC50 for the effect on growth and reproduction was reduced [100]. A few studies also noted that calculating the EC50 using water-extracted Zn and CaCl_2_-exchangeable Zn allowed them to compare the toxicity values derived in aged/leached and freshly spiked soils [88]. Nikolic et al. investigated whether the metal porewater concentration may be more appropriate for forecasting the acute toxicity of metal in collembolans due to the implications of food exposure not becoming apparent when the exposure time was short. Furthermore, as previously stated, the sensitivity of springtails is influenced by changes in soil physicochemical parameters as well as test settings [103]. As a result, many studies have found that, while the above-mentioned bioavailable fractions did connect metal toxicity to collembolans, they are not totally appropriate or satisfactory to predict metal toxicity directly. When toxicity values were calculated using water-soluble Ni, water-soluble Zn, water-soluble Cd, porewater, and CaCl_2_-exchangeable Cd, porewater Pb, CaCl_2_-exchangeable Pb, and CaCl_2_-exchangeable Zn it suggested that these metal fractions were not the only ones that collembolan species could use. Furthermore, the toxicity disparities between aged/leached and freshly spiked soils were not reduced when metal toxicity levels were represented as water-soluble Ni and porewater Pb [24,78,81].

### 4.3. Aging Time and Leaching

When conducting metal toxicity testing, solutions of the metals are typically applied to the soil. Toxic stress is invariably produced in newly spiked urban soil by high bioavailable metal concentrations, increased salinity (SO_4_^2−^, Cl^−^, NO_3_^−^, K^+^, and Na^+^), and altered pH (all of which may impact metal partitioning) [47,88,106]. To avoid overestimating metal toxicity, it is vital to leach freshly spiked soil with water and age the soil for a period of time, allowing the metal to equilibrate in the soil. Both the redox state and the bioavailable fractions will change during the aging process in soil for Pb and Cd with more than one oxidation state [89,103]. A few researchers have explored and investigated the effects of leaching and aging on the toxicity thresholds of Cd and Pb on collembolan species. Gestel et al. reported that to eliminate the excess Cl^−^, the Cd-spiked soils were leached with approximately two pore volumes of deionized water [100]. Notwithstanding the decrease in Pb concentrations in urban soil solution and the minor increase in pH in leached soil, the leaching treatment caused no noticeable variations in Pb toxicity [107].

## 5. The Use of Biochar as a Response to Urban Soil Pollution

### 5.1. Biochar Production

The two key elements influencing the properties of biochar are the preparation temperature and the feedstock. Hypothetically, the production of biochar by plants that fix carbon dioxide and then transform it into a solid residue for use in reclaiming contaminated urban soil could be a means of capturing atmospheric carbon. In terms of water filtration, soil water retention, ionic exchange, nutrient retention, and nitrogen utilization efficiency, biochar’s ultimate application to urban soil can be quite beneficial [20,44,108]. Biochar is made from a carbon-rich organic substance (such as biomass) that has been pyrolyzed at a high temperature under oxygen-restricted circumstances [44,47,109,110]. Because of the apparent heat emitted during pyrolysis, which may be turned into energy and used as a heating system for buildings, and other facilities, biochar has potential applications [111,112]. As demonstrated in Appendix A, pyrolysis is a target for operating off-gas volatilization, and a moderately modern preparation known as hydrothermal carbonization (HTC) of biomass, in which the biomass is treated with hot compressed water rather than drying, could be an alternative to slow pyrolysis [44,108,113]. In some aspects, it can demonstrate a bacterial predominance. When compared to biochar made at the same working temperature, it contains less alkaline earth and heavy metals and has a better heating value [44]. A contradiction exists between slow and fast pyrolysis. The system is specifically referred to as “slow” when slow pyrolysis conditions exist, the batch process is controllable at low temperatures, and residence time and heating rate can be lengthy and low, respectively. As a result of increased heating, aliphatic bonds in biomass are converted to aromatic bonds in biochar, resulting in chemical bond alteration. Pyrolysis starts with the destruction of hemicellulose between 200 and 260 °C, then the decomposition of cellulose and lignin between 240 and 350 °C and 280 to 500 °C, respectively [112,114].

The best methods for managing on-site waste regeneration allow for the creation of biochar from a range of waste sources, which benefits the environment. Because of this, using biochar has considerable economic and environmental advantages over the labor-intensive modification processes required to create activated carbon [114,115,116]. Additionally, biomass such as wood chips, crop waste, and animal manure are thought of as feedstock sources for the formation of biochar (Appendix A). The treatment of contaminated urban soil is another area where biochar is clearly advantageous. A variety of biomass can be used to create biochar, and its production enhances the environmental remediation of urban soil, making it a low-cost and highly effective strategy [108,115]. These potential benefits, combined with the fact that biochar made from a variety of biomass can potentially be a relatively cost-effective and environmentally friendly tool for environmental remediation, have sparked an increase in biochar research.

### 5.2. Response of Biochar Application to Collembolan Species

Both the physical and chemical characteristics of biochar may influence how it affects collembolan species. The use of biochar in small amounts to control the distribution and diversity of collembolans is growing in popularity, but unintended changes to collembolan species as a result of biochar use are also a cause for concern [116]. This is a crucial area of study because collembolan population diversity and health are crucial for ecosystem services and soil function, which in turn affect soil stability, nutrient cycling, aeration, water usage effectiveness, disease resistance, and C storage capacity [48,116,117]. This aspect suggests that organic amendments are one of the most significant tools for regulating soil biodiversity. The main reason for applying biochar is to reduce the toxicity of contaminants by adsorbing the ions on its surface [97]. When biochar enters the soil, it could interact with collembolans and facilitate their growth by reducing the acidity of the soil, which limits the growth of collembolans [118,119]. On the other hand, the contamination of biochar particles would be eaten by collembolans, which would then be poisoned by these biochar particles, and then the Pb and Cd would be transferred from the soil to higher trophic levels, such as the collembolan’s predators, via the soil food web [116]. Due to properties such as porous structures, large surface areas, and the presence of functional groups, biochar exhibits good characteristics as a sorbent or amendment for contaminants in contaminated urban soil [112,116,120]. It can have a different effect on the mobility of metals in urban soils compared to that in water [121]. As illustrated in Figure 3, changing the pH of the urban soil to increase or induce ionic exchange, surface adsorption, and potential precipitation, as well as enhance urban soil characteristics can explain mechanistic routes for metal immobilization using biochar [44,47,122]. Biochar is said to have enough surface area due to the extensive dispersion of micro- or mesoporous materials [119]. Herein, increases in the number of micropores can increase the surface area and surface sites on which metals can quickly attach [121,123]. According to many published studies [115], the mechanism of metal removal can be summarized in two terms: surface adsorption and partition. Adsorption refers to the surface interactions that cause contaminant molecules to attach to biochar surfaces, whereas sorption includes both surface adsorption and the partitioning of contaminant molecules in biochar micropores, as shown in Figure 3. Because partition occurs in the uncarbonized fraction and surface adsorption occurs in the carbonized fraction, many sorption isotherms and sorption kinetics based on batch experiments indicate that the sorption of inorganic contaminants to biochar occurs at low temperatures primarily due to partition and at high temperatures due to surface adsorption [44,46,75]. Luoz et al. reported that the sorption mechanism of sugarcane straw biochar at 700 °C was four times greater than that of 400 °C biochar due to the increased surface area at elevated temperatures [44]. For instance, the ability of biochar to complex and electrostatically interact with heavy metals is the cause for the removal of Cr, where contamination of Cd and Pb is controlled by cation exchange and precipitation, while contamination of Hg is controlled by reduction and complexation [115].

Furthermore, the sorption capacity of biochar is dependent on feedstock qualities for parameter control during the preparation process in order to create a high-quality material with the required attributes [98,122,124,125,126,127]. Although the study focused on the biochar adsorption process, there are still uncertainties about the overall mechanism due to competing impacts from other existing contaminants. For that reason, advanced technologies to improve or empower biochar with high selectivity for targeted heavy metal adsorption over a variety of other contaminants found in urban soils can be developed.

## 6. Conclusion and Future Perspective

In this review, we provided an overview of Pb and Cd contamination in the urban soil to highlight their toxicity to the collembolan species as well as their interaction with biochar and reasons for its effectiveness as a technology useful in Pb and Cd remediation/removal. This review demonstrates that soils contaminated by Pb and Cd have a negative effect not only on single species but also on collembolan species. At a national scale, the levels and risks of Pb and Cd pollution were observed to be low to medium in Chinese urban soils. On the other hand, European, American, and Asian cities had medium to high levels of Pb and Cd pollution in urban soils. Heavy metal contamination in the urban soils of many Chinese cities is influenced by the types of industrialization and urbanization histories. Furthermore, special attention should be paid in China and other investigated countries to urban greenspaces that have the potential to affect human health. Simultaneously, the direct threats posed to collembolan species and humans by current emissions of contaminants in urban soils remain a critical issue for future investigation. Additionally, the use of biochar is preferred over other sorbents not only because of its attractive properties (efficiency, affordability, and environmental friendliness) in the remediation of heavy metals, but also because it is beneficial for collembolan species through the supply of soluble carbon (C) and nutrients to collembolans. In acidic soil, biochar application can be toxic or detrimental to collembolan species. Finally, from the review, future research should focus on the preparation of biochar with desired and improved properties for further applications. We also encourage more studies on the effects of biochar on collembolan species and to provide a more comprehensive evaluation of biochar from multidimensional perspectives to see whether biochar may or may not harm collembolans.

## Figures and Tables

**Figure 1 ijerph-20-03077-f001:**
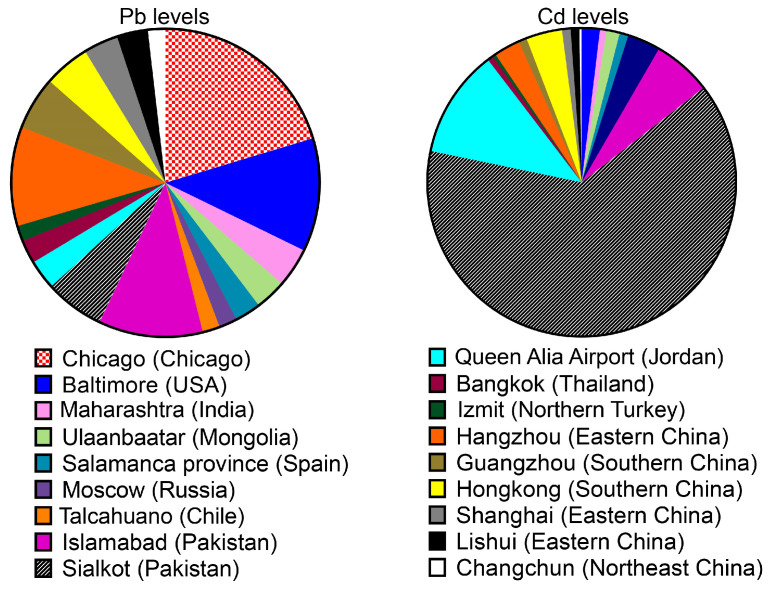
Pb and Cd levels in urban soils of Chinese cities and other cities around the world have been detected. Data are extracted from previous publications.

**Figure 2 ijerph-20-03077-f002:**
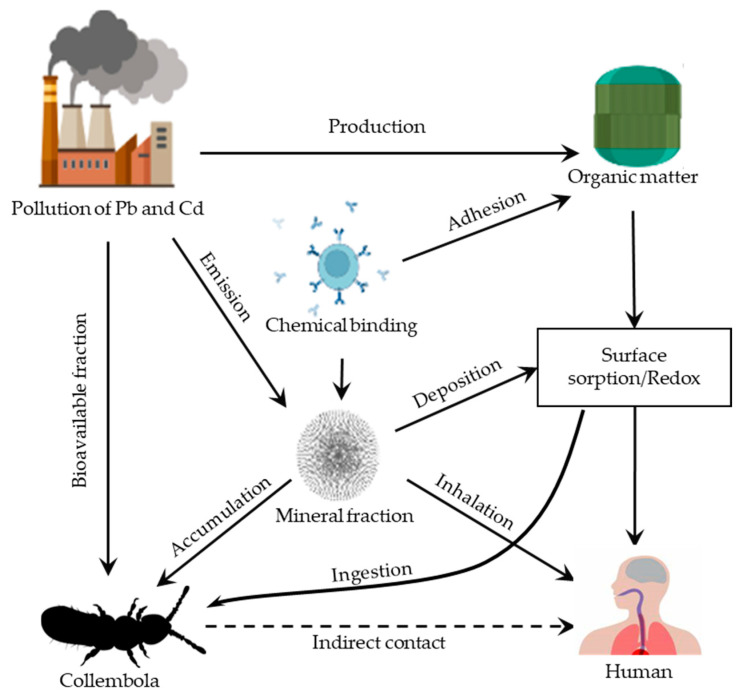
The process involved in the interactions between the Pb- and Cd-contaminated urban soil and collembolan species.

**Figure 3 ijerph-20-03077-f003:**
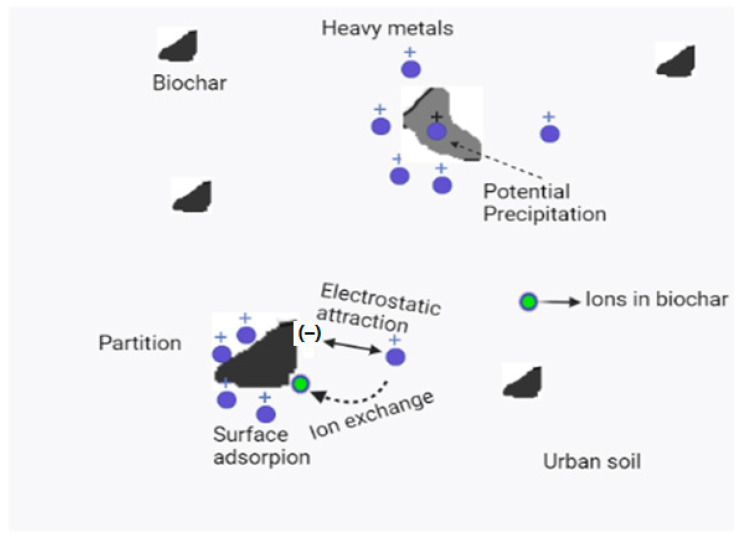
Mechanistic pathways for Pb and Cd immobilization using biochar.

## Data Availability

Not applicable.

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
