# Peer review of "The Effects of Heavy Metal Pollution on Collembola in Urban Soils and Associated Recovery Using Biochar Remediation: A Review"

_ijerph, 2023, doi:10.3390/ijerph20043077_

Round 1
Reviewer 1 Report
The manuscript entitled ''The effects of heavy metal pollution on Collembola in urban soils and associated recovery using biochar remediation: A review'' is a good work about heavy metal contamination and its immobilizing possibility using biochar. It's well-written and potentially can be published but there are some points that need to be corrected. The background of the research is good but in the biochar section, there is some room need to be filled. Some mentioned references are not related to biochar or its characteristics and need to be rechecked. The reference style also needs to be rechecked.
Please find highlights and comments in the PDF.
Good Luck!

Author Response
Response to Reviewer 1 Comments
The manuscript entitled ''The effects of heavy metal pollution on Collembola in urban soils and associated recovery using biochar remediation: A review'' is a good work about heavy metal contamination and its immobilizing possibility using biochar. It's well-written and potentially can be published but there are some points that need to be corrected. The background of the research is good but in the biochar section, there is some room need to be filled. Some mentioned references are not related to biochar or its characteristics and need to be rechecked. The reference style also needs to be rechecked.
Dear Dr.,
We appreciate for the awesome comments. We agreed with you that there is some room that needs to be filled. Please find below the detailed point-by-point response to your comments (as you highlighted in the manuscript), the responses are written in red.
Point 1. Check the font across the text, put a space between numbers and replace “parent material” with feedstock!
Response 1: Done. Please see line 115, 121,136,143, 271, 273-274, 280, 291, 304, 311 and 430
Point 2. Bring a reference for this “Best methods for managing on-site waste regeneration allow for the creation of biochar from a range of waste sources, which benefits the environment. Because of this, using biochar has considerable economic and environmental advantages over the labor-intensive modification processes required to create activated carbon”
Response 2: We have added the reference as you suggested. See Line 450-453.
Point 3. Bring this reference “https://doi.org/10.3390/plants11111424” for “The main reason for applying biochar is to reduce the toxicity of contaminants by adsorbing the ions on its surface”
Response 3: Thank you for the suggestion., we have added the suggested reference. See Line 469-470.
Point 4. I checked these references but it's not about biochar or its surface area and characteristic. Please check it again!
Response 4: The reference has been improved. See Line 477
Point 5. Add this reference: https://doi.org/10.3390/agronomy12092106 for this sentence “Biochar is said to have enough surface area due to the extensive dispersion of micro- or mesoporous materials”
Response 5: Done. (Line 481-484).
Point 6. Ref 99 is not about biochar adsorption characteristics! “Adsorption refers to the surface interactions that cause contaminants' molecules to attach to biochar surfaces, whereas sorption includes both surface adsorption and the partitioning of contaminants' molecules in biochar micropores, as shown in Fig. 5. Because partition occurs in the uncarbonized fraction and surface adsorption occurs in the carbonized fraction, many sorption isotherms and sorption kinetics based on batch experiments indicate that the sorption of inorganic contaminants to biochar occurs at low temperatures primarily due to partition and at high temperatures due to surface adsorption [99]”
Response 6: Done. (Line 485-492).
Point 7. Add author contributions, funding, institutional review board statement, informed consent statement, data availability statement, and conflicts of interest.
Response 7: Done. (Line 528-547).
Point 8. Reference style is NOT correct and should be rechecked based on instruction for author. For example, authors name should be present completely NOT with et al. Or journal names should be abbreviated in italic.
Response 8: We have improved all references and ensure all of them are in the standard Nature format and follow the sequence.
Yours sincerely,
Bin Wang

Reviewer 2 Report
The manuscript "The effects of heavy metal pollution on Collembola in urban 2 soils and associated recovery using biochar remediation: A review" addresses the important aspect of environmental pollution and its debilitating consequences to living organisms. The manuscript was well organized and the different sections are well connected.
However, there are some comments to address before publication.
1. Check the pdf file for general comments and highlights
2. The authors should avoid starting sentences with... [ref]...as it disturbs reading. Always start such sentences with ...John et al. [ref]...
3. The title suggests that biochar is an important part of the review. However, I don't think the authors considered this. There are many studies on the effect of biochar's properties on heavy metals immobilization as well as soil and other parameters. The authors should review this aspects and give weight to the title.

Author Response
Response to Reviewer 2 Comments
The manuscript "The effects of heavy metal pollution on Collembola in urban 2 soils and associated recovery using biochar remediation: A review" addresses the important aspect of environmental pollution and its debilitating consequences to living organisms. The manuscript was well organized and the different sections are well connected.
However, there are some comments to address before publication.
Dear Dr.,
We appreciate for the awesome comments. We agreed with you that there is some room that needs to be filled. Please find below the detailed point-by-point response to your comments (as you highlighted in the manuscript), the responses are written in red.
Point 1. Change centuries to countries and resulting crop failure to resulting in a decrease in agricultural productivity
Response 1: We have improved these sentences, see line 51 and 74
Point 2. We have revised “Urban soils are generally considered a complex terrestrial ecosystem because of their variety of physicochemical properties. Because soils in urban areas are generally refilled and transported from nearby systems, lake settlings, deep soils, and even construction residues” To ... "Urban soils are generally considered a complex terrestrial ecosystem because of their variety of physicochemical properties. This is because soils in urban areas are generally refilled and are transported from nearby systems, lake settlings, deep soils, and even construction residues."
Response 2: We have improved these sentences, see line 60-63.
Point 3. Add space, add a comma ",", Superscript (-1), and Add "The", Revise “6.1. and 6.2, Change to "adsorbing", Remove the 400 ºC, convert all the units to mg/kg for easy comparison with data from China, Change “[82] discovered” To "It was discovered that...” and “absorbing” to "adsorbing"
Response 3: Done. Line 155, 162,169-170,205-208, 302, 303, 304, 355,364, 428, 461, and 493
Point 4. Please separate “In addition, Menta et al. (2006) investigated the effects of Pb and Cd on the biology (survival and mortality) of two euedaphic collembolan species, F. candida and C. caeca, as well as differences in sensitivity between the two species; thus, the two species were used in this study as the indicator species, and the soil concentrations of both elements used in this study were 10, 50, 100, 500, and 1000 µg g-1 in dry soil, but no tests using both metals in one treatment were carried out” to two sentences and delete the sentence “As mentioned above, regarding other soil invertebrates, F. candida is once again one of the most sensitive taxa [49, 86, 87]”
Response 4: Done. Line 289-294.The suggested sentences has been deleted
Point 5. I think this figure is not needed. It has the same message as Figure 2
Response 5: Figure 2 has been deleted. Thank you for the suggestion.
Point 6. Revise “[89, 96, 97], the absence of rivalry with H+ for metal binding increased with increasing soil pH, which was attributable to the absence of competition with H+ for metal binding at high soil pH”, and “It is becoming more popular to use biochar in low concentrations to manage the distribution and diversity of collembolan, but accidental changes to collembolan species as a result of biochar use are of equally concern”
Response 6: Done. Line 358-360, and 463-465
Point 7. Avoid starting with reference numbers
Response 7: Done. Line 379-380, 390, L394, L399, and 420
Point 8. Revise “On the other hand, the contamination of biochar particles would be eaten by collembolans, which would then be poisoned by these biochar particles, and then the Pb and Cd would be transferred from the soil to higher trophic levels, such as the collembolan’s predators, via the soil food web”
Response 8: We have carefully checked and ensure is well written.
Point 9. Add author contributions, funding, institutional review board statement, informed consent statement, data availability statement, and conflicts of interest.
Response 9: Done. (Line 528-547).
Point 10. Reference style is NOT correct and should be rechecked based on instruction for author. For example, authors name should be present completely NOT with et al. Or journal names should be abbreviated in italic.
Response 10: We have improved all references and ensure all of them are in the standard Nature format and follow the sequence.
Yours sincerely,
Bin Wang
Reviewer 3 Report
From my point of view, the authors have worked on a very interesting topic such as the study of Collembola species present in the soil as indicators of soil health and, specifically in this case, as indicators of the presence of heavy metals in soils.
I attach the manuscript with some formatting corrections highlighted in yellow.
From the point of view of the use of biochar in urban soil bioremediation, I believe that the article includes a limited number of references because a more extensive review of the interaction between springtails and heavy metals is made.
Perhaps it would be interesting to include some information on the interactions of biota and biochar depending on the type of biomass used, since correct references have been inserted related to it.
I believe that an important issue regarding the use of biochar has been left out, and that is the risk of producing dust that can be inhaled by people, an issue that cannot be ignored when dealing with urban areas and not agricultural landscapes.
Some references to consider could be:
http://doi.org/10.1007/s13399-020-01013-4
http://doi.org/10.1016/j.scitotenv.2019.05.007
http://doi.org/10.1016/j.atmosenv.2015.10.070
https://doi.org/10.3390/su132111871

Author Response
Response to Reviewer 3 Comments
From my point of view, the authors have worked on a very interesting topic such as the study of Collembola species present in the soil as indicators of soil health and, specifically in this case, as indicators of the presence of heavy metals in soils.
Dear Dr.,
We appreciate for the awesome comments. We agreed with you that there is some room that needs to be filled. Please find below the detailed point-by-point response to your comments (as you highlighted in the manuscript), the responses are written in red.
Point 1. May be the unit m2 is not the correct unit?
Response 1: We have checked carefully the unit and found that it is correct
Point 2. Check the font across the text... I highlighted some “fungi or humus”, Put a space between numbers, add comma, and check the unit (0, 300, 600, 1200, 2400, and 4800 mg kg1), and add "The"
Response2: Done. Now we have improved the “fungi or humus” and “Folsomia candida and Coecobrya caeca”. Please see line 115,155,162, 171,184,204-208,297,304, and 355
Point 3. Fig. 1 before Table S1 (since it is within the principal document). Figure 1 and, in general, all figures and tables in the document should appear in the text just after citation.
Response 3: Done. Line 132,264, and 330, Figure3 has been deleted as suggested by one of the reviewers.
Point 4. In general, in this referencing situation, it is correct to cite the name of the author.
Response 4: Done. Line 366-367,379-380, 390, 394, 399, 420, and 492.
Point 5. From the point of view of the use of biochar in urban soil bioremediation, I believe that the article includes a limited number of references because a more extensive review of the interaction between springtails and heavy metals is made. Perhaps it would be interesting to include some information on the interactions of biota and biochar depending on the type of biomass used, since correct references have been inserted related to it. I believe that an important issue regarding the use of biochar has been left out, and that is the risk of producing dust that can be inhaled by people, an issue that cannot be ignored when dealing with urban areas and not agricultural landscapes. Some references to consider could be:
http://doi.org/10.1007/s13399-020-01013-4, http://doi.org/10.1016/j.scitotenv.2019.05.007, http://doi.org/10.1016/j.atmosenv.2015.10.070, https://doi.org/10.3390/su132111871
Response 5: Done. We really appreciate you for your time for reviewing our manuscript and providing the important comments. We have added the suggested publications in our manuscript.
Point 6. Add author contributions, funding, institutional review board statement, informed consent statement, data availability statement, and conflicts of interest.
Response 6: Done. (Line 528-547).
Point 7. Reference style is NOT correct and should be rechecked based on instruction for author. For example, authors name should be present completely NOT with et al. Or journal names should be abbreviated in italic.
Response 7: We have improved all references and ensure all of them are in the standard Nature format and follow the sequence.
Yours sincerely,
Bin Wang
Round 2
Reviewer 2 Report
The authors have revised the manuscript as suggested.